# Plasma Polyamines Decrease in Patients with Obstructive Cholecystitis

**Amaar A. Akbaraliev [1], Leila Akhvlediani [1], Ana Kavazashvili [1], Emzar Diasamidze [2,3], Omar Surmanidze [4], Nils C. Gassen [5] and Elmira A. Anderzhanova [1,5,*]**

[1] School of Medicine, BAU International University Batumi, Fridon Khalvashi St. 237, 6010 Batumi, Georgia
[2] Republican Clinical Hospital, Tbel Aburseridze St. 2, 6010 Batumi, Georgia
[3] Batumi Medical Center (BMC), Kakhaberi St. 36, 6010 Batumi, Georgia
[4] Health Center "Medina", Fridon Khalvashi St. 237, 6010 Batumi, Georgia
[5] Clinic of Psychiatry and Psychotherapy, University Hospital Bonn, Venusberg-Campus 1, 53127 Bonn, Germany
[*] Correspondence: elmira.anderzhanova@googlemail.com

**Abstract:** Polyamines (PAs), endogenous metabolites with a wide range of biological activities, are synthesized at a high rate in liver supporting hepatocyte proliferation and survival. The liver appears as an important regulator of plasma PAs; however, the perspective to exploit plasma PA measurements as indicators for liver function was not explored. This study aimed to evaluate the value of the plasma levels of PAs as a biomarker of pathological changes in the liver in patients with obstructive cholecystitis. The levels of polyamines and their acetylated forms were measured using HPLC/UV in the plasma of patients with obstructive cholecystitis and in healthy subjects. PA turnover was assessed by the ratio between an acetylated form of PA and PA. An effect of diet preference of cheese or meat, the major exogenous sources of PAs, smoking, and severe acute respiratory syndrome coronavirus 2 (SARS-CoV-2) in anamnesis was also evaluated in healthy subjects. We found that the plasma levels of spermine and acetylated spermidine decreased in patients with obstructive cholecystitis without a concurring increase in the total plasma bilirubin and amylase levels. The turnover of spermine and spermidine was also changed, suggesting a decrease in the rate of PA degradation in the liver. In healthy subjects, the PA levels tended to mirror chronic smoking and recent SARS-CoV-2 infection but were not relevant to diet factors. A number of observations indicated the role of physical exercise in the regulation of the plasma pool of PA. The decrease in plasma PA levels and index of PA turnover in the cholestasis syndrome indicate the liver's metabolic function reduction. A conceivable effect of lung-related conditions on plasma PA, while indicating low specificity, nonetheless, speaks favorably about the high sensitivity of plasma PA measurement as an early diagnostic test in the clinic.

**Keywords:** biomarker; cholecystectomy; cholecystitis; polyamines; polyamine metabolism; SARS-CoV-2

## 1. Introduction

The need for more sensitive and specific biomarkers of diseases is actualized nowadays. It is encouraged by a quest for early and individual-based diagnostics and fostered by a better understanding of the pathophysiology of disorders.

Acute and chronic cholecystitis is an inflammatory disease of the gallbladder with a prevalence of 10–20% [1]. Up to 90% of acute and chronic cholecystitis are associated with gallstones in the gallbladder. Chronic cholecystitis is usually asymptomatic and most frequently diagnosed clinically during an acute exacerbation. Functional diagnostics using ultrasound may not be sensitive, while the more specific cholescintigraphy (HIDA-scan) is expensive in diagnosing cholecystitis [2]. Therefore, an early laboratory diagnostic is highly anticipated. Known laboratory tests for liver dysfunction, such as total and direct bilirubin, alanine aminotransferase (ALT)/aspartate aminotransferase (AST) [3,4], as well

as amylase and lipase measurements (to exclude obstruction of the common pancreatic duct by gallstones) are well-validated but not optimal measures showing 80% sensitivity and 75% specificity for the diagnosis of cholecystitis. In turn, they are not fair enough to monitor early pathological changes in the liver, such as nonspecific reactive hepatitis, which are induced by an obstruction of the bile flow [5].

Polyamines (PAs) are endogenously synthesized, biologically active compounds involved in the regulation of growth, proliferation, and maintenance of cells of different origins. PAs putrescine (PTR), spermidine (SPD), and spermine (SPM) are of highest biological significance. The synthesis and turnover of PTR, SPD, and SMP are tightly controlled by specific enzymes, such as, ornithine decarboxylase, SPD and SPM synthases (SDS, SMS), and SPD/SPM acetyltransferase (SAT1), which are highly expressed in the liver [6].

PAs mediate plenty of vital functions in hepatocytes. Recently, it was proposed that PAs are strong epigenetic regulators of autophagy; therefore, are essential to maintain protein homeostasis in the living cell [7]. PAs also promote protein synthesis in the liver [8] and are required for liver regeneration in the condition of experimental partial hepatectomy [9,10]. SPD and SPM were found to regulate the hepatocyte growth factor (HGF)—mediated DNA synthesis and counteract targeted suppression of its activity [11].

Levels of PA and PA metabolism are important indicators of neoplastic changes, also in the liver [12], and biomarkers of an efficient anticancer therapy [13]. Nonetheless, it is still a question whether, how, and at which conditions PAs exert a protective effect. Changes in PAs levels appear as important components of the physiological stress response [14–16]; therefore, PA may have a protective effect. PAs were found to increase levels of anti-inflammatory cytokines and suppress lymphocyte activity and neutrophil migration to inflamed tissues. Exogenous PTR, SPD, and SPM were found to decrease inflammation: they decrease lipid peroxidation in the liver that appears together with a decline in the AST and ALT levels [17].

These pieces of evidence suggest that the biochemical parameters of the PA system are a sensitive measure to assess the functional activity of the liver. It is of interest to explore if PA measurements in the peripheral blood can be utilized as both diagnostic and prognostic biomarkers of liver dysfunction in clinical hepatology. The present study's goal was to evaluate the biomarker potential of PA measurements in blood by analyzing plasma PTR, SPD, and SPM and their acetylated metabolites N1-acetylSPD (NAcSPD) N1-acetylSPM (NAcSPM) in patients with obstructive cholecystitis.

## 2. Materials and Methods

### 2.1. Design of the Study

The study was a screening study, designed as a prospective observational analytical controlled cross-sectional double-blinded mono-centered investigation.

### 2.2. Demographic Data

Patients with obstructive cholecystitis ($n = 22$) were compared to healthy controls ($n = 27$). Ages of subjects were between 20 and 82 years (obstructive cholecystitis) and between 19 and 63 years (healthy individuals). Genders were presented equally and χ-square test showed no difference in thegender proportion between groups of healthy subjects and patients) (df = 3.302, z = 1.817, $p = 0.0692$). All subjects were white Caucasians: Georgian, Slavic, and Turkic. Every participant was given a 1-page form with questions regarding his/her age, sex, weight/height, diet, aged-cheese consumption, and history of SARS-CoV-2 exposure and consequences. Aged-cheese consumption dietary factor was specified because several studies showed increased polyamine levels in people who consumed more aged-cheeses in their diet. Body mass indexes (BMI) were calculated based on actual measurements.

### 2.3. Surgical Intervention

Laparoscopic cholecystectomy was performed under general anesthesia with propofol for induction and propofol and isoflurane mix for support using a standard protocol.

### 2.4. Blood Collection and Sample Preparation

In patients with cholecystitis, blood samples were collected before (S1) and on the 6th or 7th 3rd postoperative day (S2) in the morning after an overnight fasting. In healthy subjects, S1 and S2 samples were collected, keeping 3–4 days in between. Blood was centrifuged at RT and $2000\times g$ for 15 min. Plasma was frozen immediately and kept at $-20\,^{\circ}\text{C}$ prior to analysis.

### 2.5. Polyamine Measures and Analysis

Chemicals and organic solvents of an analytical or HPLC-grade were from Carl Roth, GmbH, Karlsruhe, Germany. Dansyl chloride (Cat. N. 39220) and standards, putrescine dihydrochloride (Cat. N. S5780) spermidine trihydrochloride (Cat. N. A2501), and spermine tetrahydrochloride (Cat. N. S2876) were from Sigma-Aldrich, Hamburg, Germany.

Heparinized plasma (200 µL) was mixed with 200 µL of 5% trichloracetic acid (TCA) (the dilution ratio is 1:2) and centrifuged at $3500\times g$ at room temperature (RT) for 10 min. The supernatant (250 µL) was withdrawn and mixed, sequentially, with 50 µL of NaOH 2M, 75 µL of saturated $NaHCO_3$, and 500 µL of dansyl chloride (10mg/mL in 100% acetone). The mix was incubated at $60\,^{\circ}\text{C}$ for 60 min. Dansylation of target analytes was stopped by adding 50 µL of $NH_4OH$ 28% and subsequent incubation at $60\,^{\circ}\text{C}$ for 20 min. Five hundred microliters of reaction mixture was mixed with 200 µL of 100% acetonitrile and centrifuged at $11{,}000\times g$ at RT for 10 min for polyamine extraction. The upper phase was used for HPLC analysis. Pas and their metabolites were measured using high-pressure liquid chromatography with ultraviolet detection (HPLC-UV) [18,19]. Separation was performed, as in [20], with adaptation using the Ultimate3000 system (Thermo-Fisher Scientific, Dreieich Germany) under a binary gradient (water pH = 4.0 and acetonitrile 100%) at a constant flow rate of 0.35 mL/ min and at the temperature of $32\,^{\circ}\text{C}$. Eluent A was HPLC-grade water (pH = 4, adjusted with 85% $H_3PO_4$) and eluent B was 100% acetonitrile. The flow gradient condition was as follows: 0–3 min 50% B; 20–26 min 90% B; 27–40 min 50% B with linear changes of eluent partial composition. Detection of eluted analytes was performed by measurement of light absorption at 254 nm with a data collection rate of 10 Hz. The injection volume was 25 µL. Quantification of Pas was performed using the external calibration method. Concentration absorbance dependence was linear within the selected range of concentrations (1–10,000 ng/mL). Limits of detection/quantification were from 0.35 to 0.52 ng/mL.

The rate of production of SPD and SPM was estimated by evaluation of SPD/PTR and SPM/PTR ratios that allows one to indirectly assess the activity of spermidine synthase (SRM) and sum activity of SRM and spermine synthase (SMS). The spermine/spermidine N1-acetylytransferase-1 (SAT1) activity toward either SPM or SPD was evaluated as NAc-SPD/SPD and NAcSPM/SPM, respectively [21].

### 2.6. Biochemical Measurements

Total bilirubin (TB) and amylase plasma levels were measured by standard colorimetric assays (AMY-P 20766223 322; BILT3 05795397 190, Roche Diagnostics GmbH, Hongkong) in accordance to the manufacturer's instructions.

### 2.7. Statistical Analysis

The sample size calculations were performed using GPower 3.1 software (Franz Faul, Heinrich-Heine-Universität Düsseldorf, Düsseldorf, Germany) and were performed based on estimated Cohen's 1.2 < d <1.5. Data (mean $\pm$ SEM) were checked for outliers using the ROUT protocol, Q = 1% [22], and further analyzed using the GraphPad Prism 8.0 software (GraphPad Software, San Diego, CA, USA). Comparisons of means were performed using

the two-tailed unpaired Student's *t*-test. Two-way mixed effect ANOVAs with subsequent post hoc tests, when appropriate, were performed to examine the effects of pathology or time of collection (the two primary factors of analysis). Pearson's correlation analysis was performed to analyze the correlations. *p*-values < 0.05 were considered statistically significant.

## 3. Results

### 3.1. Basic Clinical and Biochemical Data Do Not Show the Characteristic Endophenotype in Patients with Obstructive Cholecystitis

BMI did not differ between the disease group (33.2) and control group (26.4) (Student's two-tailed unpaired *t*-test, t = 1.951, df = 45, *p* = 0.057). Analysis revealed a significant correlation between age and BMI in groups of healthy ($R^2$ = 0.27, *p* = 0.006) but not diseased ($R^2$ = 0.06, *p* = 0.28) individuals (Figure 1).

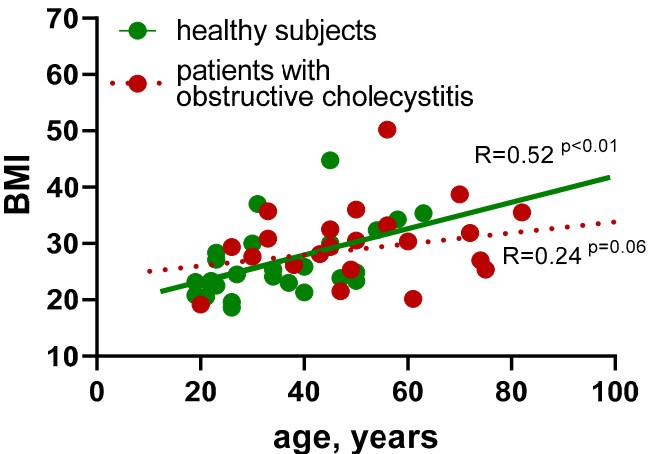

**Figure 1.** Correlations between age and body mass indexes (BMI) in patients with obstructive cholecystitis and in healthy subjects.

Biochemical measurements of total bilirubin (TB) and amylase (Amyl) plasma levels were performed in patients only. Most of the data were within the normal range (<50 U/L for Amyl and <20 μmol/L for TB). TB levels showed positive correlation with amylase levels ($R^2$ = 0.89, *p* < 0.001) (Figure 2). The very high correlation is achieved due to one patient with very high levels of Amyl and TB (125.5 U/L and 52.2 mM/L, respectively). These data were not considered outliers, but rather were kept since even without this point, correlation is still significant ($R^2$ = 0.43, *p* = 0.013). TB levels also tended to negatively correlate with NAcSPM levels ($R^2$ = 0.12, *p* = 0.1) and SPM/PTR ratio ($R^2$ = 0.06, *p* = 0.26) (data are not shown).

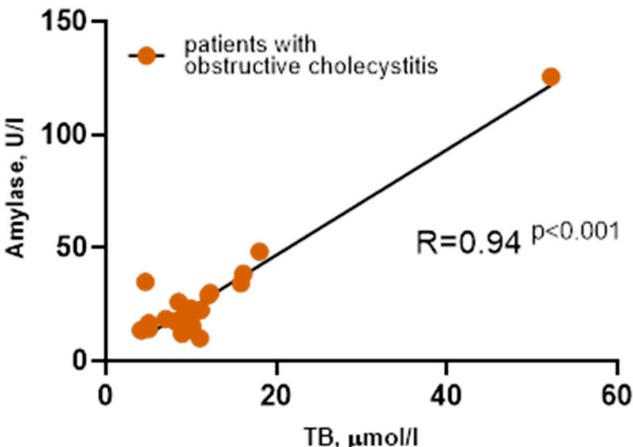

**Figure 2.** Correlation between plasma total bilirubin and amylase levels.

### 3.2. Obstructive Cholecystitis Leads to the Decrease in SPM and NAcSPD in Blood

We revealed small bidirectional effects of time on the majority of plasma Pas and their metabolites (two-way ANOVA: pathology $\times$ sample, Fs > 4.54, ps < 0.4), except NAcSPM (F (1,31) = 0.18, ps = 0.668), which were remaining stable (Figure 3E).

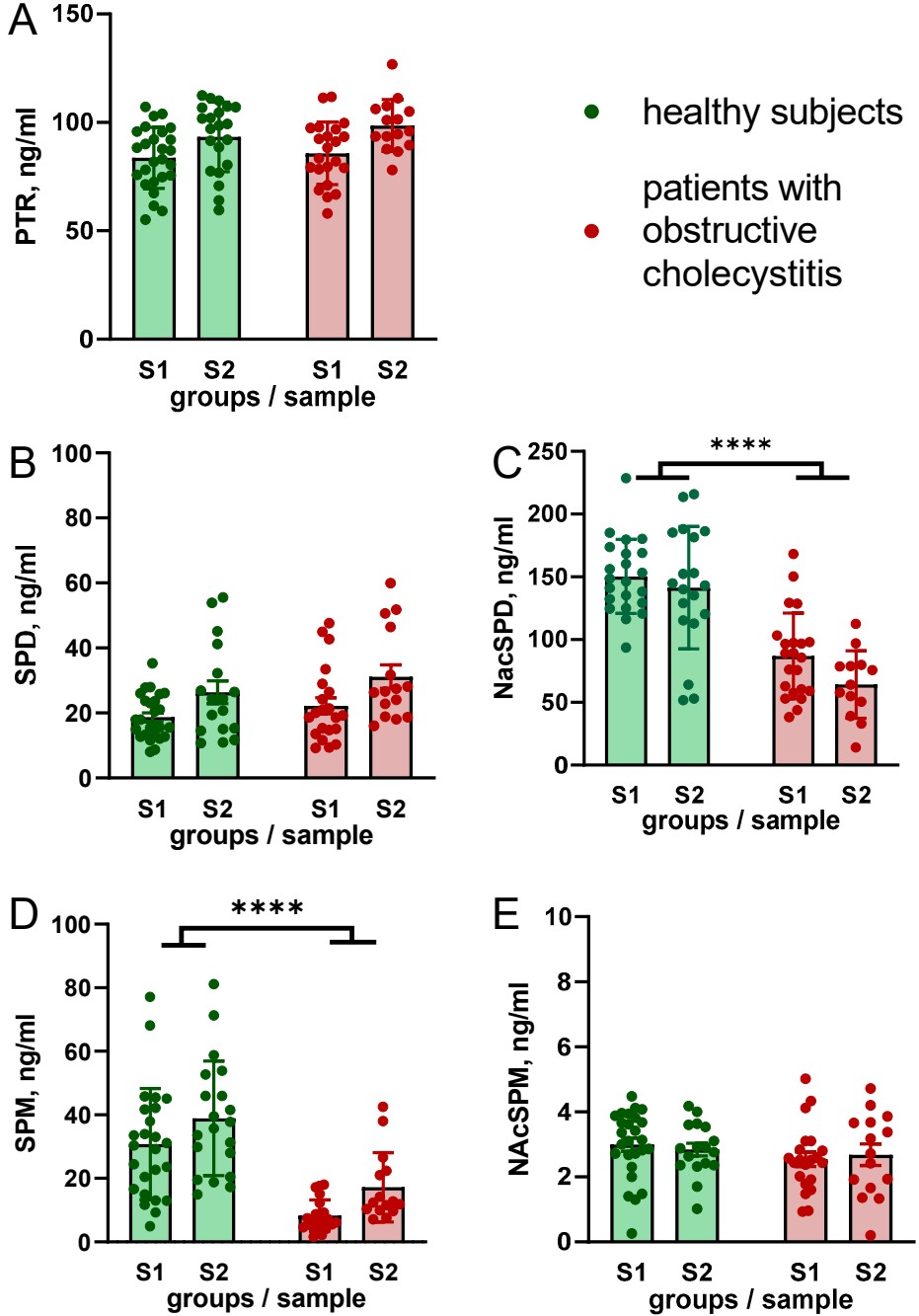

**Figure 3.** Profile of plasma polyamines in healthy subjects and patients with cholecystitis and liver cirrhosis. Comparison of plasma levels of (**A**) putrescine (PTR); (**B**) spermidine (SPD); (**C**) $N^1$Ac spermidine (NAcSPD); (**D**) spermine (SPM); (**E**) $N^1$Ac spermine (NAcSPM) in healthy and patients with obstructive cholecystitis. S1—Before cholecystectomy/first blood sample; S2—after cholecystectomy/second blood sample. ****—$p$ < 0.0001.

Obstructive cholecystitis appeared as a significant factor changing NAcSPD and SMP plasma levels (two-way ANOVA: pathology $\times$ sample, F (1,46) = 62.64, $p$ < 0.0001, Figure 3C)

and SPM (F (1,46) = 31.25, $p < 0.0001$, Figure 3D). Neither PTR, SPD, nor NAcSPM levels were affected (Fs < 1.72, ps > 0.192).

### 3.3. PA Turnover Is Changed in Patients with Obstructive Cholecystitis

We assessed PA turnover, comparing ratios between concentrations of respective PAs in S1 plasma samples of diseased and healthy subjects. Student's two-tailed unpaired *t*-test showed that SPD/PTR ratios did not differ (t = 0.954, df = 45, $p = 0.345$, Figure 4A), while SPM/PTR ratios did differ between groups (t = 5.577, df = 44, $p < 0.001$, Figure 4B). Student's two-tailed unpaired *t*-test showed a significant difference between NAcSPD/SPD indexes between groups (t = 5.294, df = 38, $p < 0.001$, Figure 4C), therefore, suggesting a decrease in SAT1 activity is obstructive cholecystitis. In contrast, the SPM to NAcSPM transformation rate was increased (t = 5.316, df = 41, $p < 0.001$, Figure 4D). However, the two-time difference between SPM and SPD levels (Figure 1B,D) might lead to this apparent increase being due to a lower substrate availability.

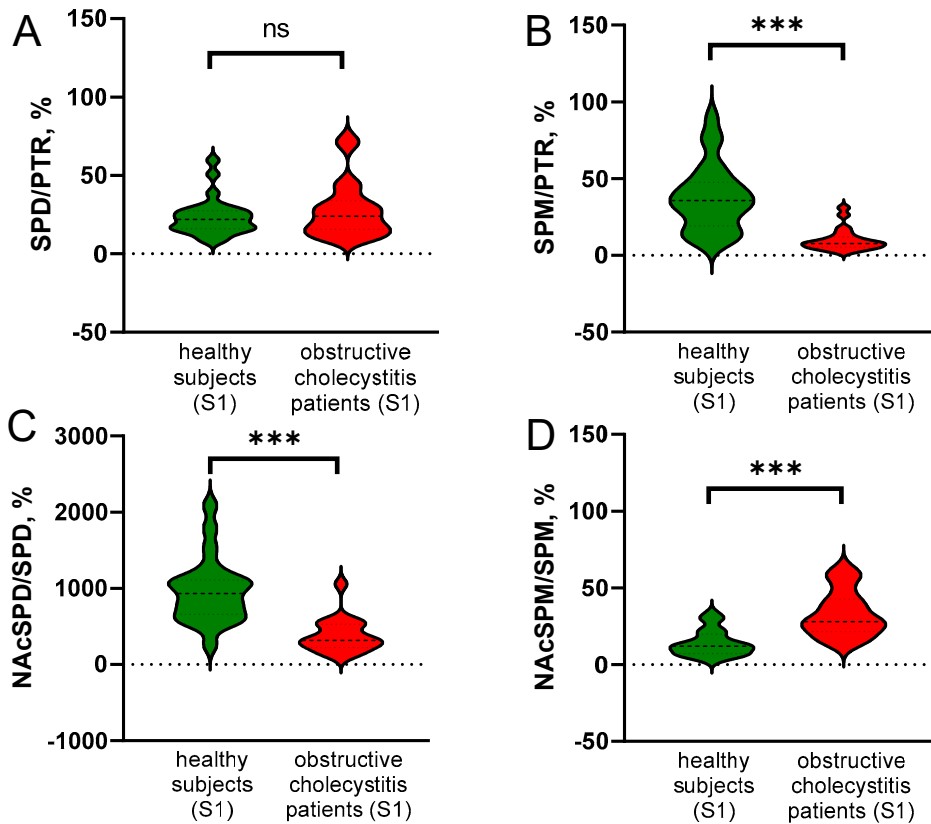

**Figure 4.** Estimated rate of production of PAs in healthy subjects and patients with cholecystitis. Approximate rates of conversion of (**A**) putrescine (PTR) to spermidine (SPD); (**B**) PTR to spermine (SPM); (**C**) SPD to N1Ac spermidine (NAcSPD); (**D**) SPM to N1Ac spermidine (NAcSPM). ***—$p < 0.001$; ns—not significant.

Analysis of all paired correlations between all measured and calculated parameters of the PA system revealed the negative correlation between PTR and SPD levels ($R^2 = 0.20$, $p = 0.04$) and the positive correlation between PTR and NAcSPD levels ($R^2 = 0.28$, $p = 0.01$) in patients with cholecystitis.

### 3.4. Plasma SPM Levels Decreased in Healthy Subjects Recovered from SARS-CoV-2

The concentrations of polyamines in the plasma of healthy subjects (S1) were: PTR 83.7 ng/mL, SPD 18.7 ng/mL, NAcSPD 150.3 ng/mL, SPM 30.8 ng/mL, and NAcSPM 2.9 ng/mL.

The analysis of the influence of dietary factors (cheese or meat preference), smoking, and SARS-CoV-2 infection in recent anamnesis revealed the effect of past SARS-CoV-2 on SPM plasma levels (Student's test, t = 2.066, df = 22, $p$ = 0.05,). This was in accord with the coinciding decreasing tendency of the SPM/PRT ratio (Student's test, t = 1.267, df = 23, $p$ = 0.21) and the increasing tendency of the NAcSPD/SPD ratio (Student's test, t = 1.562, df = 20, $p$ = 0.13).

Smoking also tended to decrease the SPM/PTR ratio (Student's test, t = 1.347, df = 23, $p$ = 0.19) that corresponded to increased PRT plasma levels (Student's test, t = 1.347, df = 24, $p$ = 0.18).

None of the two dietary factors under investigation did influence PA and PA metabolite levels in healthy subjects (Student's test, ts < 1.062, ps > 0.31).

### 3.5. Candidate Regulatory Factors of Plasma PAs

Assuming a significant decrease in PA levels in patients with obstructive cholecystitis, we selected healthy individuals in whom two or three parameters measured were fulfilling the Q1 inclusion criteria (fitted to the lowest quartile of the entire data pull). We found that such healthy individuals (*n* = 5) had a few additional factors, which may influence PA levels (Table 1). Surprisingly, regular physical exercise or healthy weight loss coincided with the decrease in PA levels.

**Table 1.** Healthy subjects with low PA levels (within Q1 range) and their concurring or medical conditions.

| Individual's ID | SPM (<15.8 ng/mL) | SPM/PTR(<19.1%) | NAcSPD/SPD (<657.2%) | Additional Medical Conditions |
|---|---|---|---|---|
| 30 | 13,060 | 16,224 | | regular crossfit training |
| 31 | 4940 | 8949 | 547,891 | weight loss 12 kg/2 months |
| 32 | 12,920 | 14,629 | 547,287 | viral hepatitis C—recovered/ regular weightlifting |
| 74 | | | 265,006 | combined oral contraceptive use |
| 75 | 11,720 | 10,933 | | using proton pump inhibitors for peptic ulcer disease |

### 4. Discussion and Conclusions

This is the first study reporting the changes in plasma PAs levels in patients with obstructive cholecystitis and showing the decrease in PA synthesis and metabolism in diseased individuals in comparison to healthy subjects (Figures 3–5). This decrease was significant even under the pressure of such factors as smoking and SARS-CoV-2 in anamnesis, which were emerging as regulators of PA plasma levels in healthy individuals.

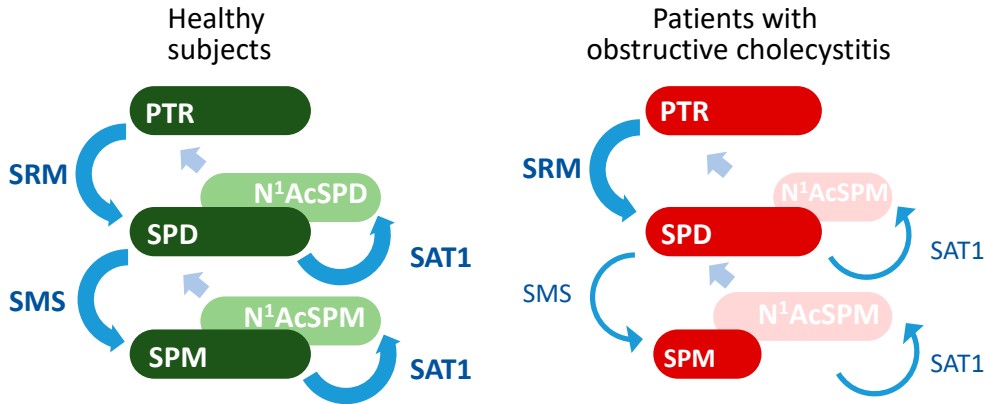

**Figure 5.** Graphical summary: The proposed difference in the metabolism of PA in healthy subjects and in patients with obstructive cholecystitis. Abbreviation as in Figures 2 and 3 legends.

Published data suggest a tight relationship between the PA biochemical system and hepatocyte vitality. As was shown, the hepatoprotective effect of anthocyanins in a model of carbon tetrachloride-induced toxicity was associated with the decrease in the tissue PTR and SPM decay via oxidation [23] that speaks in favor of PA as an integrative indicator of liver function. The significant decrease in the total plasma levels of SPM as well as in the SAT1-dependent acetylation of SPD in patients with obstructive cholecystitis (Figure 3D) indicate a suppression in activity of PA metabolizing enzymes, particularly, SMS and SAT1 (Figure 4B–D). However, the conclusion on the decrease in SAT1 activity is compromised by the apparent increase in the conversion of SPM to its acetylated metabolite. This phenomenon can be caused by the substrate's wane: the residual activity of SAT1 may be sufficient to convert the available SPM into its acetylated metabolite. These observations suggest a possible sustainable deterioration of liver metabolic function, which is developing upon chronic cholestatic syndrome. Thus, we did not observe any positive dynamics in PA levels upon successful cholecystectomy within 3–4 days after surgery. The decrease in parameters of the PA biochemical system was in a strict contrast to bilirubin and amylase measurements, which were within normal ranges (Figure 2). Therefore, our results suggest the PA measurements as a sensitive biomarker to monitor liver malfunctions.

Detailed analysis of the data on the plasma polyamine levels in healthy subjects revealed a few interesting results, which are further stressing the fact of a high sensitivity of the polyamine system to physiological stress. Firstly, the PA levels mirrored chronic smoking and recent SARS-CoV-2 infection. It is known that both SARS-CoV-2 and smoking provoke inflammation in liver tissues [24,25]. Therefore, PA metabolism may be changed due to changes in the functional activity of hepatocytes. However, changes in PA levels may be directly related to the state of the lung, and the observed decrease in PA levels might be mediated by the inflammation in the pulmonary tissue [26]. The exact causality of the observed phenomena is of undoubtful interest, but it is out of the scope of the present study.

We expected that plasma PA levels will be reflective of a diet in healthy subjects, but we did not observe any statistical difference in the plasma PA comparing between individuals with preferential cheese/meat and preferential vegetarian diets. We can offer at least two explanations of dissolution of the expected effect of diets. We did not stratify subjects under investigation toward a strict adherence to vegetarian food; also, other confounding factors (such as, smoking and SARS-CoV-2) may contribute to the higher variability in PA plasma levels.

Our case observations indicated a role of physical exercise in the regulation of plasma PA levels that also suggest a high sensitivity of the PA system to physiological stress. The observed decrease in plasma SPM in individuals performing regular physical exercise may be mediated by changes in PA metabolism in skeletal muscles related to SPM oxidase overexpression occurring in trained individuals [27]. In turn, the apparent decrease in SPM levels in sportsmen may be related to a fast turnover of PAs within the PTR-SPD-SPM-SPD loop. SPD, which is of high metabolic demand in muscles in regularly trained personalities, was proposed to be an important regulator of skeletal muscle atrophy/hypertrophy. Specifically, regulation of autophagy may be a candidate mechanism of beneficiary action of PA in muscles. The combination of exercise and exogenous SPD induces autophagy in myocytes through the activation of the AMPK-FOXO3a signal pathway and is suppressed by changes in the expression of antiapoptotic Bcl-2 and proapoptotic Bax and caspase-3 [28].

This study has a significant limitation related to the fact that only one pathological condition was taken for consideration. It can be speculated that changes in PA levels may have a biphasic character, depending on both the type of liver pathology and/or stage of the disease. Moreover, the changes in PA plasma levels may occur due to bile toxicity and represent changes in the PA metabolism in the periphery. The study merits to be continued also to check the aforementioned possible pitfalls.

To conclude, the decrease in plasma PA levels and index of PA turnover in cholestasis syndrome are indicative of the possible decline in the liver's metabolic function (Figure 5).

Plasma PA measurements appear as a sensitive indicator of liver pathology; however, its prognostic value may be limited due to their low specificity.

**Author Contributions:** A.A.A.: original draft preparation, samples' preparation, data analysis, reviewing, and editing; L.A.: sample preparation, reviewing, and editing; A.K.: sample preparation and analysis; E.D.: supervision and data curation; O.S.: data curation and reviewing; N.C.G.: conceptualization and reviewing; E.A.A.: supervision, conceptualization, methodology, data analysis, reviewing, and editing. All authors have read and agreed to the published version of the manuscript.

**Funding:** The study was performed with functional support from BAU University Batumi, Georgia.

**Institutional Review Board Statement:** All procedures were approved by the Ethics Committee of BAU Batumi International University (approval code N-EMD004) and met the requirement of the Declaration of Helsinki for Medical Research Involving Human Subjects [29].

**Informed Consent Statement:** Informed consent was obtained from all subjects involved in the study. Written informed consent has been obtained from the patient(s) to publish this paper. Every participant was introduced to the reasons, procedures, benefits, and possible side effects of the study and had to fill and sign a special form of informed consent provided in both the Georgian and English languages. Any participant was able to cancel his consent at any time of the study.

**Data Availability Statement:** Data is contained within the article.

**Acknowledgments:** The authors wish to thank Nevsha Tajelipirbazari, BAU Batumi Student Self Government, the medical and administrative staff of Batumi Medical Center, Health Center "Medina", and Republican Clinical Hospital for their support.

**Conflicts of Interest:** The authors declare no conflict of interest.

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
