# Peer review of "Plasma Polyamines Decrease in Patients with Obstructive Cholecystitis"

_livers, doi:10.3390/livers2030019_

Round 1
Reviewer 1 Report
This study is interesting to consider polyamines as biomarkers of the functional state of the liver in health and disease. The results are promising in that there are differences between healthy groups and those with obstructive cholecystitis.The English is excellent. Considering its relevance and originality, I recommend a minor revision. Please see specific comments below.
Minor concerns:
1) Why fig 5 appears at the beginning?, it would be convenient to number the figures in the order of appearance.
2) In methodology 2.5 paragraph, 3500 G change to 3500 g and 60 C change to 60ºC.
2) Figure 3 and Figure 4, check names of series and x-axis change Sublets to subjets and obstrictive to obstructive
3) Fig 4 What does ns refer to in the first plot?
4) In Ethical considerations, they mentions the approval by the ethics committee but does not provide the number of the permit granted by the committee.
Author Response
Authors thank Reviewer for a high evaluation of the present study. The minor revision was done in accordance to suggestions. Changes and insets are highlighted in yellow.
- Why fig 5 appears at the beginning? it would be convenient to number the figures in the order of appearance.
We agree that such an order is rather confusing therefore we removed the figure reference in the Introduction and put is in a more suitable pace in the discussion.
- In methodology 2.5 paragraph, 3500 G change to 3500 g and 60 C change to 60ºC.
Changed in accordance.
- Figure 3 and Figure 4, check names of series and x-axis change Sublets to subjets and obstrictive to obstructive
Changed in accordance.\
- Fig 4 What does ns refer to in the first plot?
ns - not significant
The respective clarification was added in the figure’s legend.
- In Ethical considerations, they mentions the approval by the ethics committee but does not provide the number of the permit granted by the committee.
The requested information was added into the Materials and Method section.
Reviewer 2 Report
This is a descriptive but interesting paper on changes in polyamine levels during and after cholecystitis. A few issues need to be resolved.
Patient values should be in the main manuscript. This should include things such as ALT, bilirubin, AST, GGT, total bile acids, any instances of cirrhosis, etc… Preferably this would also include some sort of inflammatory status like a CBC. This is an important and necessary point. The authors should also note the time frame under which these assays occurred. Were all patients undergoing acute cholecystitis? Did all patients have ALT increases? ALP increases?
The incredibly strong R value for amylase and bilirubin indicates there may be some sort of contaminating overlap issue with the basic assay. Did the authors try another kit to validate these findings? This could help alleviate concerns the kits are reacting with the samples themselves.
It may be useful to compare these clinical data to animal models, both for basic science researchers and for clarifying potential mechanisms.
Author Response
Authors appreciate Reviewer for the critical remarks, allowing us to improve the manuscript. Changes and insets are highlighted in yellow.
Patient values should be in the main manuscript. This should include things such as ALT, bilirubin, AST, GGT, total bile acids, any instances of cirrhosis, etc… Preferably this would also include some sort of inflammatory status like a CBC. This is an important and necessary point. The authors should also note the time frame under which these assays occurred. Were all patients undergoing acute cholecystitis? Did all patients have ALT increases? ALP increases?
We do understand that the very best scenario should include a wider biochemical profiling of the peripheral markers of biochemical cytolytic and inflammatory syndromes relevant to liver function. ALT and AST measurement definitely would add reliability to our conclusion on impact of hepatocyte vitality on PA metabolism. However, changes in PA metabolism and/or their abnormal secretion may be even more early markers of changes in hepatocyte homeostasis and around-apoptosis metabolite secretome that would increase the value of PA measurements in clinical practice. ALT, AST and ALP were measures in not all patients, only in case of a necessity to clarify a degree of hepatocellular syndrome severity and to be better prepared for post-operational treatment. Therefore, the entire pool of these biochemical data was not statistically representative to report in the present paper? therefore was not included.
All patients were individuals with chronic cholecystitis with repeatedly verified clinical diagnosis including US examination.
We appreciate the suggestion to estimate the inflammatory markers, and we are palming to measure the metabolome in the next subset of patients with obstructive cholecystitis to have a better view of low molecular weight biomarkers of the pathological state.
The incredibly strong R value for amylase and bilirubin indicates there may be some sort of contaminating overlap issue with the basic assay. Did the authors try another kit to validate these findings? This could help alleviate concerns the kits are reacting with the samples themselves.
Honestly saying, we do not understand what exactly stands behind the term “contaminating overlap” and would highly appreciate Reviewer for clarifying the claim. However, we believe that the issue may be related to the problem of signal saturation due to high optical density of solution in case of high concentration of analytes and respective ceiling effect, which may be affecting spectrometric measurement of Amyl and TB.
The very high correlation we report in the manuscript was achieved also due to one patient with very high levels of Amyl and TB ((125.5 U/l and 52.2 mM/l, respectively)/ we did not consider these data as outliers, rather were keeping them since even without this point correlation is still significant (p = 0.013). We add a comment in Result section.
Assuming the stimulating role of bilirubin on amylase secretion (Hirohata et al., 2002) and association between obstruction of common bile duct and pancreatitis (Güngör et al., 2011) we consider correlation between plasma levels of total bilirubin and amylase pathophysiologicaly relevant. Therefore, we do not presume any contamination issue related to quality of assay.
We also would like to stress that the biochemical kits used in the study are from a well-recognized manufacturer and approved for clinical studies. Validity of the test is controlled by withing laboratory and between laboratories quality controls and also confirmed by a relevance of the data to actual clinical condition of patients.
Güngör, B., CaÄŸlayan, K., Polat, C., Seren, D., Erzurumlu, K., Malazgirt, Z., 2011. The predictivity of serum biochemical markers in acute biliary pancreatitis. ISRN Gastroenterol 2011, 279607. https://doi.org/10.5402/2011/279607
Hirohata, Y., Fujii, M., Okabayashi, Y., Nagashio, Y., Tashiro, M., Imoto, I., Akiyama, T., Otsuki, M., 2002. Stimulatory effects of bilirubin on amylase release from isolated rat pancreatic acini. American Journal of Physiology-Gastrointestinal and Liver Physiology 282, G249–G256. https://doi.org/10.1152/ajpgi.00429.2000
It may be useful to compare these clinical data to animal models, both for basic science researchers and for clarifying potential mechanisms.
We do appreciate reviewer for this suggestion. Firstly, we believe that we need to accumulate more data which would bring more sight on the pathological mechanisms involved. Beside planned metabolomic study we also are going to measure peripheral autophagy markers assuming signaling fiction of PA.
Based on the present data, we rather limited in speculation. The reports on changes in plasma PA with respect to liver function are mostly done considering hyperplastic changes in the liver, therefore related biochemical and molecular biological mechanisms might be different. Nonetheless, we did our best to incorporate key publications in the field of a direct relevance to our original study.